# DiffMimic: Efficient Motion Mimicking with Differentiable Physics

**Jiawei Ren**[*1]  **Cunjun Yu**[*2]  **Siwei Chen**[2]  **Xiao Ma**[3]  **Liang Pan**[1]  **Ziwei Liu**[1]

[1] S-Lab, Nanyang Technological University
[2] School of Computing, National University of Singapore
[3] SEA AI Lab

## Abstract

Motion mimicking is a foundational task in physics-based character animation. However, most existing motion mimicking methods are built upon reinforcement learning (RL) and suffer from heavy reward engineering, high variance, and slow convergence with hard explorations. Specifically, they usually take tens of hours or even days of training to mimic a simple motion sequence, resulting in poor scalability. In this work, we leverage differentiable physics simulators (DPS) and propose an efficient motion mimicking method dubbed **DiffMimic**. Our key insight is that DPS casts a complex policy learning task to a much simpler state matching problem. In particular, DPS learns a stable policy by analytical gradients with ground-truth physical priors hence leading to significantly faster and stabler convergence than RL-based methods. Moreover, to escape from local optima, we utilize an *Demonstration Replay* mechanism to enable stable gradient backpropagation in a long horizon. Extensive experiments on standard benchmarks show that DiffMimic has a better sample efficiency and time efficiency than existing methods (*e.g.*, DeepMimic). Notably, DiffMimic allows a physically simulated character to learn Backflip after 10 minutes of training and be able to cycle it after 3 hours of training, while the existing approach may require about a day of training to cycle Backflip. More importantly, we hope DiffMimic can benefit more differentiable animation systems with techniques like differentiable clothes simulation in future research. [1] [2]

## 1 Introduction

Motion mimicking aims to find a policy to generate control signals for recovering demonstrated motion trajectories, which plays a fundamental role in physics-based character animation, and also serves as a prerequisite for many applications such as control stylization and skill composition. Although tremendous progress in motion mimicking has been witnessed in recent years, existing methods (Peng et al., 2018a; 2021) mostly adopt reinforcement learning (RL) schemes, which require alternatively learning a reward function and a control policy. Consequently, RL-based methods often take tens of hours or even days to imitate one single motion sequence, making their scalability notoriously challenging. In addition, RL-based motion mimicking highly relies on the quality of its designed  (Peng et al., 2018a) or learned (Peng et al., 2021) reward functions, which further burdens its generalization for complex real-world applications.

Recently, differential physics simulator (DPS) has achieved impressive results in many research fields, such as robot control (Xu et al., 2022) and graphics (Li et al., 2022). Specifically, DPS treats physics operators as differentiable computational graphs, and therefore gradients from objectives (*i.e.*, rewards) can be directly propagated through the environment dynamics to control policy functions. Instead of alternatively learning between reward functions and control policies, the control policy learning tasks can be resolved in a straightforward and efficient optimization manner with

---

[*] Equal contribution, listed in alphabetical order.
[1] Our code is available at https://github.com/diffmimic/diffmimic.
[2] Qualitative results can be viewed at https://diffmimic-demo-main-g7h0i8.streamlitapp.com/.

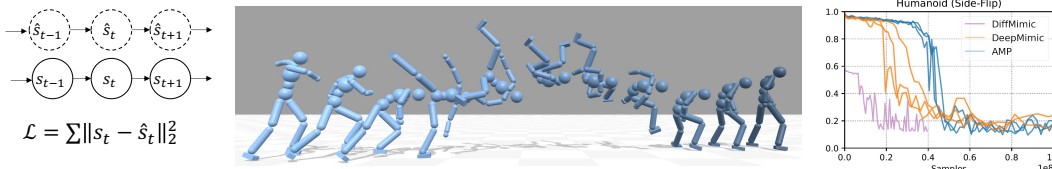

Figure 1: Overview of our method. **Left:** DiffMimic formulates motion mimicking as a straight-forward state matching problem and uses analytical gradients to optimize it with off-the-shelf differentiable physics simulators. The formulation results in a simple optimization objective compared to heavy reward engineering in RL-based methods. **Middle:** DiffMimic is able to mimic highly dynamic skills, *e.g.*, Side-Flip. **Right:** DiffMimic has a significantly better sample efficiency and time efficiency than state-of-the-art motion mimicking methods. Our approach usually achieves high-quality motion (pose error $< 0.15$ meter) using less than $2 \times 10^7$ samples.

the help of DPS. However, despite their analytical environment gradients, optimization with DPS could easily get into local optima, particularly in contact-rich physical systems that often yield stiff and discontinuous gradients (Freeman et al., 2021; Suh et al., 2022; Zhong et al., 2022). Besides, numerical gradients could also vanish/explode along the backward path for long trajectories.

In this work, we propose DiffMimic, a fast and stable motion mimicking method with the help of DPS. Different from RL-based methods that require heavy reward engineering and poor sample efficiency, DiffMimic reformulates motion mimicking as a state matching problem, which could directly minimize the distance between a rollout trajectory generated by the current learning policy and the demonstrated trajectory. Thanks to the differentiable DPS dynamics, gradients of the trajectory distance can be directly propagated to optimize the control policy. As a result, DiffMimic could significantly improve the sample efficiency with the first-order gradients.

However, simply utilizing DPS could not guarantee global optimal solutions. In particular, the roll-out trajectory tends to gradually deviate from the expert demonstration and could produce a large accumulative error for long motion sequences, due to the distributional shift between the learning policy and expert policy. To address these problems, we introduce the *Demonstration Replay* training strategy, which randomly inserts reference states into the rollout trajectory as anchor states to guide the exploration of the policy. Empirically, Demonstration Replay gives a smoother gradient estimation, which significantly stabilizes the policy learning of DiffMimic.

To the best of our knowledge, DiffMimic is the first to utilize DPS for motion mimicking. We show that DiffMimic outperforms several commonly used RL-based methods for motion mimicking on a variety of tasks with high accuracy, stability, and efficiency. In particular, DiffMimic allows learning a challenging *Backflip* motion in only 10 minutes on a single V100 GPU. In addition, we release the DiffMimic simulator as a standard benchmark to encourage future research for motion mimicking.

## 2 RELATED WORK

**Motion Mimicking.** Motion mimicking is a technique used to produce realistic animations in physics-based characters by learning skills from motion captures (Hong et al., 2019; Peng et al., 2018a;b; Lee et al., 2019). This approach has been applied to various downstream tasks in physics-based animation, including generating new motions by recombining primitive actions (Peng et al., 2019; Luo et al., 2020), achieving specific goals (Bergamin et al., 2019; Park et al., 2019; Peng et al., 2021), and as a pre-training method for general-purpose motor skills (Merel et al., 2018; Hasenclever et al., 2020; Peng et al., 2022; Won et al., 2022). The scalability of these tasks can be limited by the motion mimicking process, which is a key part of the pipeline in approaches like ScaDiver (Won et al., 2020). In this work, we demonstrate that the problem of scalability can be addressed using differentiable dynamics.

**Speeding Up Motion Mimicking** Most motion mimicking works are based on a DRL framework (Peng et al., 2018a; Bergamin et al., 2019), whose optimization is expensive. Several recent works speed up the DRL process by hyper-parameter searching (Yang & Yin, 2021) and constraint relaxation (Ma et al., 2021). Another line of work learns world models to achieve end-to-end gradi-

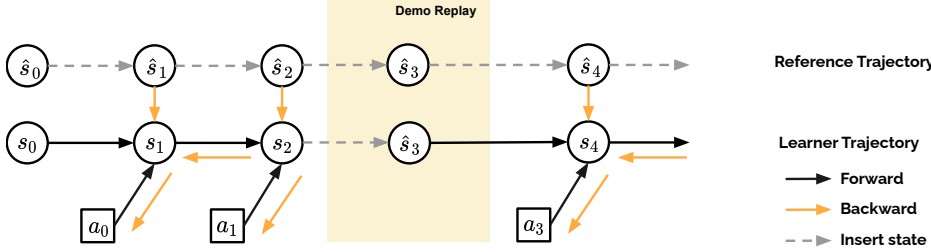

Figure 2: Computation graph of DiffMimic. An example of demonstration replay during the third step is demonstrated. The demonstration state in the third step is used to replace the simulated state.

ent optimization (Won et al., 2022; Fussell et al., 2021). However, learning world models requires extra training and introduces the risk of error accumulation. We are the first to use the off-the-shelf DPS for motion mimicking, which achieves better sample efficiency than optimized DRL frameworks and requires no additional world models.

**Differentiable Physics.** Differentiable physics gains traction recently with the emerging differentiable physics simulators (DPS) (Hu et al., 2019; 2020; Huang et al., 2021; Qiao et al., 2021). It has brought success in many domains and provides a distinctly new approach to control tasks. Supertrack (Fussell et al., 2021) proposes to learn a world model to approximate the differentiable physics and has achieved promising results. However, its policy horizon is limited during training due to the accumulation of errors in the world model. Various applications stem from DPS like system identification (Jatavallabhula et al., 2021), continuous controls (Lin et al., 2022; Xu et al., 2022) in the past few years and demonstrate promising results. Compared to black-box counterparts which learn the dynamics with neural networks (Fussell et al., 2021; Li et al., 2020), differentiable simulations utilize physical models to provide more reliable gradients with better interpretability. On the other hand, the analytical gradient from DPS suffers from noise or can sometimes be wrong (Zhong et al., 2022; Suh et al., 2022) in a contact-rich environment. Non-smoothness or discontinuity in the contact event requires specific techniques to compute the gradient. However, they often introduce noise into the gradient (Zhong et al., 2022). Thus, in this work, we empirically show how to handle the noise in the gradient in a contact-rich environment for motion mimicking.

**Policy Optimization with Differentiable Physics Simulators.** The integration of differentiable simulators in policy optimization has been a significant advancement in reinforcement learning. With analytical gradient calculation, the sample efficiency of policy learning is significantly improved (Mozer, 1989; Mora et al., 2021; Xu et al., 2022). However, the implementation is challenged by the presence of noisy gradients and the risk of exploding or vanishing gradients (Degrave et al., 2017). The policy may also get stuck at sub-optimal points due to the local nature of the gradient. To address these issues, various approaches have been proposed. SHAC (Xu et al., 2022) truncates trajectories into smaller segments to prevent exploding or vanishing gradients, but this requires careful design. On the other hand, PODS (Mora et al., 2021) utilizes second-order gradients, leading to monotonic policy improvement and faster convergence compared to first-order methods. However, it relies on strong assumptions about the second-order derivatives of the differentiable simulator and is sensitive to their accuracy. An alternative approach is the Imitation Learning framework with DPS proposed by ILD (Chen et al., 2022). Although it is simple and effective for robot control, it struggles with the exploration of highly dynamic motions. To tackle these challenges, we introduce a replay mechanism to enhance its performance.

## 3 METHODOLOGY

### 3.1 A MOTION MIMICKING ENVIRONMENT IN DIFFERENTIABLE PHYSICS ENGINE.

**Environment.** We build our environment in Brax (Freeman et al., 2021). We design our simulated character following DeepMimic (Peng et al., 2018a). The humanoid character has 13 links and 34 degrees of freedom. The character has a weight of 45 kg and a height of 1.62m. Contact is applied to all links with the floor. The environment is accelerated by GPU parallelization. The physics

simulator updates at a rate of 480 FPS. The joint limits of the character are relaxed to allow smoother gradient propagation. We keep the system configurations like friction coefficients consistent with DeepMimic.

**State and Action.** States include the position $p$, rotation $q$, linear velocity $\dot{p}$ and angular velocity $\dot{q}$ of all links in the local coordinate. Additionally, following Peng et al. (2018a), a phase variable $\phi$ in the range $[0, 1]$ is included in the state to serve as a timestamp. Thus, the state of the environment, $s$, contains the aforementioned information, $s := \{p, q, \dot{p}, \dot{q}, \phi\}$. We follow the common practice and use PD controllers to drive the character. Given the current joint angle $q$, angular velocity $\dot{q}$ and a target angle $\tilde{q}$, the torque on the actuator of the joint will be computed as:

$$\tau = k_p(\tilde{q} - q) + k_d(\dot{\tilde{q}} - \dot{q}), \tag{3.1}$$

where $\dot{\tilde{q}} = 0$, $k_p$ and $k_d$ are manually-specified gains of the PD controller. We keep $k_p$ and $k_d$ identical to the PD controller used in DeepMimic. A policy network predicts the target angle for the PD controller at each joint. The control policy operates at 30 FPS.

## 3.2 Motion Mimicking with Differentiable Physics

Motion Mimicking is about matching the policy rollout with the reference motion. While it has a straightforward objective, considering human action can be rich and highly flexible, designing a reward to incentivize and guide policy learning is not easy. For example, reward functions that are tailored to guide a policy to learn to walk or learn to backflip can be different and this makes reward engineering difficult. We reveal that the mimicking task can be surprisingly easy when analytical gradients can be obtained through DPS.

The underlying idea of DiffMimic is to allow the gradient to directly flow back from the distance between reference motion and policy rollout to the policy through an off-the-shelf DPS. We show the computation

---

**Algorithm 1** DiffMimic

1: **Input:** Optimization Iteration I, Episode Length T, Reference States $\hat{S} = \{\hat{s}_1, \hat{s}_2, \cdots, \hat{s}_t\}$, Error Threshold $\epsilon$.
2: **Output:** Optimized policy
3: Initialize the stochastic policy as $\pi_\theta$.
4: **for** optimization iteration $i = 1 \cdots I$ **do**
5:     # Roll out trajectories with Expert Replay
6:     Initialize $s_1 \leftarrow \hat{s}_1$
7:     **for** step $t = 1 \cdots T - 1$ **do**

$$s_{t+1} = \begin{cases} \mathcal{T}(s_t, a_t), \ a_t \sim \pi_\theta(a|s_t) & \text{if } \|s_t - \hat{s}_t\|_2^2 < \epsilon \\ \mathcal{T}(\hat{s}_t, a_t), \ a_t \sim \pi_\theta(a|\hat{s}_t) & \text{otherwise.} \end{cases}$$

8:     **end for**
9:     Compute loss $\mathcal{L} = \sum_{t=1}^{T} \|s_t - \hat{s}_t\|_2^2$
10:     Update the policy $\pi_\theta$ with analytical gradient $\bigtriangledown_\theta \mathcal{L}$.
11: **end for**
12: **Return** the policy $\pi_\theta$.

---

graph of DiffMimic in Fig. 2. Compared with existing RL-based approaches, DiffMimic directly optimizes the policy with supervised state-matching signals and largely eases the reward engineering. Thanks to the analytical gradient, such an approach has a very high sample efficiency. Prior works (Fussell et al., 2021) explored analytic gradients with a learned differentiable world model which describes the dynamics in latent space. However, the learned world model reveals no underlying physics of the system. Consequently, they undermine the effectiveness, generality, and interpretability of the learning with physics-based analytic gradients from DPS, as in DiffMimic. In addition, online learning and inference with learned world models remain computationally expensive when compared with DPS.

We show in detail how to train the policy with DPS. For each iteration, we initialize the state to the first reference state and then roll out to the maximum episode length in a batch of parallel environments. Then we compute the step-wise L2 distance between the rollout trajectory and the reference trajectory in terms of positions and rotations:

$$\mathcal{L} = \sum_{t=1}^{T} \|s_t - \hat{s}_t\|_2^2, \tag{3.2}$$

$$\|s_t - \hat{s}_t\|_2^2 \triangleq \frac{1}{\|J\|} \sum_{j \in J} w_p(p^j - \hat{p}^j)^2 + w_r(q^j - \hat{q}^j)^2 + w_v(\dot{p}^j - \hat{\dot{p}}^j)^2 + w_a(\dot{q}^j - \hat{\dot{q}}^j)^2,$$

where $p^j$ and $\hat{p}^j$ are the global positions of the $j$-th joint from rollout and reference, $q^j$ and $\hat{q}^j$ are the global rotation of the $j$-th joint from rollout and reference in 6D rotation representation. The weights

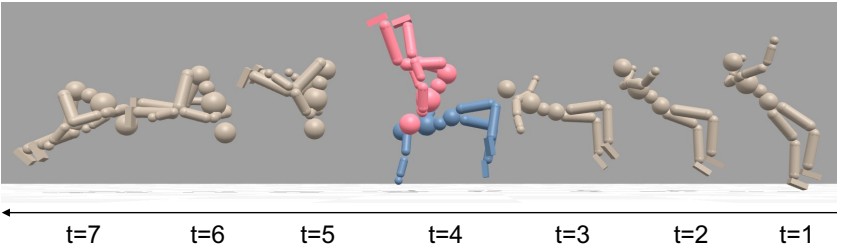

| t=7 | t=6 | t=5 | t=4 | t=3 | t=2 | t=1 |

Figure 3: Illustration of Demonstration Replay, which happens when t=4 where the character is about to fall. The rollout state (in blue) is replaced by the reference state (in red) for the next step.

$w_p$, $w_r$, $w_v$, and $w_a$ only need to be approximately tuned to roughly equalize their magnitudes. Note that we use 6D rotation in loss computation since it is advantageous over quaternions in gradient-based optimization (Zhou et al., 2019). Without loss of generality, we denote the transition function in the dynamic system as $\mathcal{T}$ and the next state of the system can be obtained from the transition function given the current action $a_t$ and character state $s_t$, $s_{t+1} = \mathcal{T}(s_t, a_t)$. In DiffMimic, DPS serves as a transition function $\mathcal{T}$ that is fully differentiable. Thus, from the loss function, we can directly derive the gradient with respect to both the current action $a_t$ and state $s_t$

$$\frac{\partial \mathcal{L}}{\partial a_t} = \left( \frac{\partial \mathcal{L}}{\partial \mathcal{T}(s_t, a_t)} \right) \left( \frac{\partial \mathcal{T}(s_t, a_t)}{\partial a_t} \right), \quad \frac{\partial \mathcal{L}}{\partial s_t} = \left( \frac{\partial \mathcal{L}}{\partial \mathcal{T}(s_t, a_t)} \right) \left( \frac{\partial \mathcal{T}(s_t, a_t)}{\partial s_t} \right). \quad (3.3)$$

By doing this recursively, the gradient can be propagated through the whole trajectory.

## 3.3 DEMONSTRATION REPLAY

Although mimicking with DPS results in a succinct learning framework, there are three well-known challenges in policy learning with DPS: 1) Exploding/vanishing gradients with the long horizon; 2) Local minima may cause gradient-based optimization methods to stall; 3) Noisy or wrong gradients.

We observe that the high non-convexity in the motion mimicking task poses severe challenges to analytical gradient-based optimization. Due to the local nature of the analytical gradients, the policy can be easily stuck at suboptimal points. As shown Tab. 6, when trying to mimic the *Backflip* skill, the character learns to use arms to support itself instead of exploring a more dynamic jumping move. In practical implementations, dividing the whole trajectory into smaller sub-trajectories for truncated Back-Propagation-Through-Time (BPTT) further exacerbates the issue. As shown in Tab. 7 (a)-(b), a 10-step gradient truncation leads the policy to a less favorable local optimal.

Teacher forcing (Williams & Zipser, 1989) is a common remedy to escape the local minimum in gradient-based sequential modeling. Teacher forcing randomly replace the states in rollout by the reference states. Given a teacher forcing ratio $\gamma$:

$$s_{t+1} = \begin{cases} \mathcal{T}(s_t, a_t), \ a_t \sim \pi_\theta(a|s_t) & \text{if } b = 0, b \sim \text{Bernoulli}(\gamma) \\ \mathcal{T}(\hat{s}_t, a_t), \ a_t \sim \pi_\theta(a|\hat{s}_t) & \text{otherwise.} \end{cases} \quad (3.4)$$

However, although the mechanism leads to a better global optimal overall, it does not guarantee the character to faithfully mimic the reference motion in each frame. As shown in Tab. 6, although the character's overall trajectory matches the reference, several frames suffer from awkward poses. Fig. 8 shows that a few frames have significantly larger pose errors than other frames.

We propose demonstration-guided exploration to mitigate these challenges in DPS for policy learning to help exploration and encourage faithfully mimicking at the same time. The main idea of demonstration-guided exploration is to replace states in the policy rollout with the corresponding demonstration states when the states are too far away from the reference. For a threshold $\epsilon$:

$$s_{t+1} = \begin{cases} \mathcal{T}(s_t, a_t), \ a_t \sim \pi_\theta(a|s_t) & \text{if } \|s_t - \hat{s}_t\|_2^2 < \epsilon \\ \mathcal{T}(\hat{s}_t, a_t), \ a_t \sim \pi_\theta(a|\hat{s}_t) & \text{otherwise.} \end{cases} \quad (3.5)$$

Since the criterion for choosing which states to replace is based on the performance of the current rollout, the replacing frequency dynamic adjusts itself during the model training.

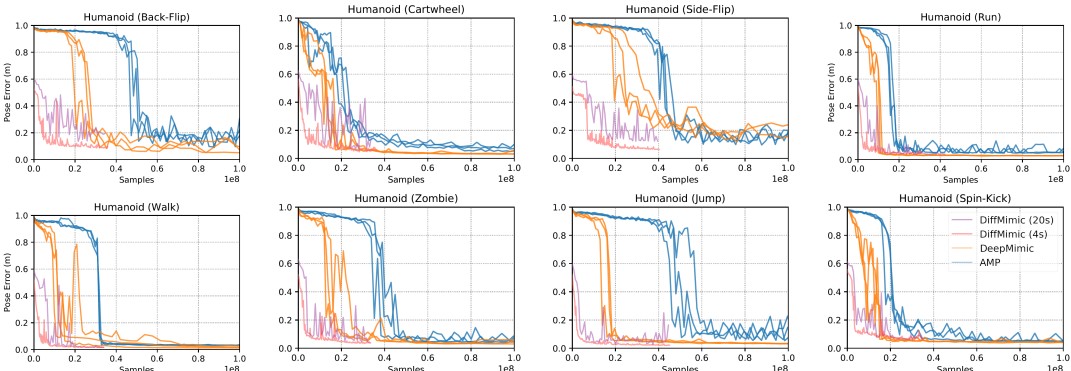

Figure 4: Pose error versus the number of samples. DiffMimic (4s): rollout 4 seconds of Diffmimic for evaluation. DiffMimic (20s): rollout 20 seconds of Diffmimic for evaluation. In general, DiffMimic allows the policy to generate high-quality motions with less than $3.5 \times 10^7$ samples. We refer the readers for more results to the appendix.

# 4 EXPERIMENTS

## 4.1 EXPERIMENT SETUP

For all the experiments, we run the algorithm with one single GPU (NVIDIA Tesla V100) and CPU (Intel Xeon E5-2680). Following Peng et al. (2021), we use the average pose error as the main metric. The average pose error over the whole trajectory of length $T$ with $J$ joints is computed between the pose of the simulated character and the reference motion using the relative positions of each joint with respect to the root joint (in units of meters):

$$e = \frac{1}{T} \sum_{t \in T} \frac{1}{\|J\|} \sum_{j \in J} \|(p_t^j - p_t^{root}) - (\hat{p}_t^j - \hat{p}_t^{root})\|_2, \tag{4.1}$$

where $p_t^j$ and $\hat{p}_t^j$ are the position of $j$-th joint of the simulated motion and reference motion at timestamp $t$ in the 3D cartesian space and $root$ refers to the root joint. We mainly compare with DeepMimic (Peng et al., 2018a), Spacetime Bound (Ma et al., 2021), and Adversarial Motion Prior (AMP) (Peng et al., 2021). Dynamic Time Warping (Sakoe & Chiba, 1978) is applied to synchronize the simulated motion and the reference motion following the convention.

**Cyclic Motions.** Besides mimicking a single motion clip, a popular benchmark for motion mimicking is to imitate cyclic motions like walking and running, where a single cycle of the motion is provided as the reference motion and the character learns to repeat the motion for 20 seconds. Typically, a curriculum is designed to gradually increase the maximum rollout episode to 20 seconds during training (Yang & Yin, 2021; Peng et al., 2018a). In our experiment, we remove all bells and whistles and fix the maximum rollout episode to 4 seconds during the training. DiffMimic can produce a 20-second long cyclic rollout even though it only sees 4-second rollouts in training. The motion clips are directly borrowed from AMP (Peng et al., 2021), which are originally collected from a combination of public mocap libraries, custom recorded mocap clips, and artist-authored keyframe animations.

## 4.2 COMPARISON ON MOTION MIMICKING

In this section, we aim to understand 1) efficiency; 2) the quality of the learned policy of DiffMimic. Following the conventions of previous works (Peng et al., 2018a; 2021; Ma et al., 2021), we count the number of samples required to train the policy to rollout for 20 seconds without falling. The pose error is calculated over the horizon of 20 seconds.

**Analytical gradients enhance the sample efficiency in motion mimicking.** We show the comparison between DiffMimic, DeepMimic, and Spacetime Bound on sample efficiency in Table 2. DeepMimic is an RL-based algorithm with a careful reward design. Spacetime Bound performs hyperparameter searching for DeepMimic to further enhance the sample efficiency. Our results

Table 1: Pose error comparison in meters. $T_{cycle}$ is the length of the reference motion for a single cycle. The error is averaged on 32 rollout episodes with a maximum length of 20 seconds.

| Motion | $T_{cycle}$(s) | DeepMimic | AMP | Ours |
|---|---|---|---|---|
| Back-Flip | 1.75 | $0.076 \pm 0.021$ | $0.150 \pm 0.028$ | $0.097 \pm 0.001$ |
| Cartwheel | 2.72 | $0.039 \pm 0.011$ | $0.067 \pm 0.014$ | $0.040 \pm 0.007$ |
| Crawl | 2.93 | $0.044 \pm 0.001$ | $0.049 \pm 0.007$ | $0.037 \pm 0.001$ |
| Dance | 1.62 | $0.038 \pm 0.001$ | $0.055 \pm 0.015$ | $0.070 \pm 0.003$ |
| Jog | 0.83 | $0.029 \pm 0.001$ | $0.056 \pm 0.001$ | $0.031 \pm 0.002$ |
| Jump | 1.77 | $0.033 \pm 0.001$ | $0.083 \pm 0.022$ | $0.025 \pm 0.000$ |
| Roll | 2.02 | $0.072 \pm 0.018$ | $0.088 \pm 0.008$ | $0.061 \pm 0.007$ |
| Run | 0.80 | $0.028 \pm 0.002$ | $0.075 \pm 0.015$ | $0.039 \pm 0.000$ |
| Side-Flip | 2.44 | $0.191 \pm 0.043$ | $0.124 \pm 0.012$ | $0.069 \pm 0.001$ |
| Spin-Kick | 1.28 | $0.042 \pm 0.001$ | $0.058 \pm 0.012$ | $0.056 \pm 0.000$ |
| Walk | 1.30 | $0.018 \pm 0.005$ | $0.030 \pm 0.001$ | $0.017 \pm 0.000$ |
| Zombie | 1.68 | $0.049 \pm 0.013$ | $0.058 \pm 0.014$ | $0.037 \pm 0.002$ |

show that DiffMimic constantly outperforms DeepMimic in terms of sample efficiency. The analytical gradients provided by the differentiable simulation allow us to compute policy gradient with a small number of samples while the RL-based algorithm requires a large batch to have a decent estimate. Compared with Spacetime Bound, DiffMimic is much more stable and consistent over various tasks. We notice that Spacetime Bound may require more samples than DeepMimic even for simple tasks like *Jump*. We show the learning curve of DiffMimic over eight different tasks in Fig. 4. It shows that DiffMimic generally learns high-quality motions (pose error $< 0.15$) with less than 2e7 samples, even for challenging tasks like *Backflip*. In terms of the wallclock time, DiffMimic learns to perform *Backflip* in 10 minutes and learns to cycle it in 3 hours ($14.88 \times 10^6$ samples), which can be found in our qualitative results.

**DiffMimic can learn high-quality motions.** The average pose error for 12 different motion tasks is shown in Table 1. DiffMimic outperforms AMP consistently and has comparable performance to DeepMimic. Noticeably, DiffMimic only needs to see the demonstrations of 4 seconds to achieve a similar performance of DeepMimic with 20 seconds cyclic rollout, which indicates stable and faithful recovery of the reference motion. This further corroborates the efficacy of DiffMimic.

**Analytical gradients accelerate the learning speed in motion mimicking.** As it generally takes more time to compute analytical gradients than the estimated one, it is natural to compare the wall-clock time for each algorithm. Since the implementation of DeepMimic does not utilize GPU for parallelization, we choose to compare DiffMimic

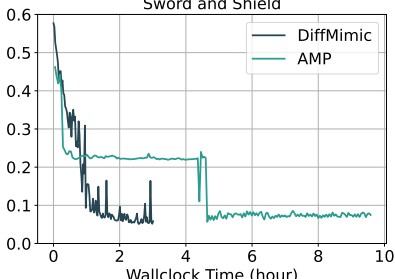

Figure 5: Wallclock training time versus pose error for DiffMimic and AMP. DiffMimic takes half of the training time required by AMP to have a comparable result.

with AMP. The implementation of AMP utilizes the highly parallelized environment, Isaac-Gym (Makoviychuk et al., 2021). Both approaches do not require manual reward design and utilize GPU for acceleration. We compare DiffMimic and AMP on a task where a humanoid character is wielding a sword and a shield to perform a 3.2-second attack move (shown in Appendix Fig. 13). The hyperparameter of AMP is from its official release of code. We show the comparison with respect to wallclock time in Fig. 5. DiffMimic takes half of the training time required by AMP to have a comparable result.

## 4.3 ABLATION ON TRUNCATION LENGTH

Training the policy with DPS would suffer from the vanishing/exploding gradients and local optimal. Recent research (Xu et al., 2022) points out that such a problem can be mitigated by a truncated learning window, which splits the entire trajectory into segments with shorter horizons. we carried out experiments to validate whether such an idea can be directly applied in motion mimicking with

Table 2: Number of samples required to roll out 20 seconds without falling in $(10^6)$. Percentage: change in the fraction of the DeepMimic samples.

| Motion | $T_{cycle}(s)$ | DeepMimic | Spacetime Bound | Ours |
|---|---|---|---|---|
| Back-Flip | 1.75 | 31.18 | 41.20 +32.1% | 14.88 -52.2% |
| Cartwheel | 2.72 | 30.45 | 17.35 -43.0% | 13.92 -54.2% |
| Walk | 1.25 | 23.80 | 4.08 -79.5% | 7.92 -66.7% |
| Run | 0.80 | 19.31 | 4.11 -78.7% | 8.16 -57.7% |
| Jump | 1.77 | 25.65 | 41.63 +77.8% | 5.28 -79.4% |
| Dance | 1.62 | 24.59 | 10.00 -59.3% | 16.56 -32.6% |

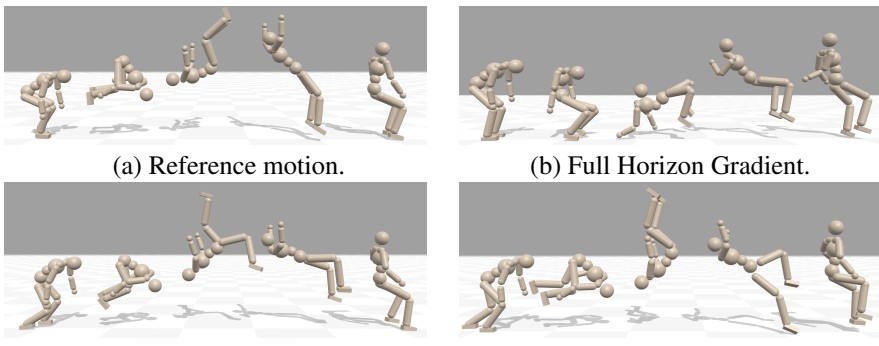

(a) Reference motion.   (b) Full Horizon Gradient.

(c) Demonstration Replay (Random).   (d) Demonstration Replay (Threshold).

Figure 6: Qualitative results of three variants of DiffMimic with respect to demonstration replay. The policy trained with full horizon gradient fails to jump up. The policy trained with Demonstration Replay (Random) fails to recover the reference motion faithfully.

DPS. Truncation length refers to the horizon over which the gradient is calculated. The quantitative results are shown in Fig. 7. Indeed, it is difficult to learn a good policy by propagating the gradient through a whole trajectory that is long. In addition, simply splitting the whole trajectory into segments would worsen the final performance. We hypothesize the naive truncation of the trajectory creates discontinuities in the whole trajectory whereas motions in the trajectory are highly interdependent. For example, how to flip in mid-air closely relates to how the character jumps. We see this as a strong call for a better strategy to handle these two challenges in motion mimicking with DPS.

## 4.4 ABLATION ON DEMONSTRATION REPLAY

To understand how demonstration replay affects the policy learning of DiffMimic, we compare three different variants of DiffMimic, namely, Demonstration Replay (Random) and Demonstration Replay (Threshold), and Full Horizon Gradient on *Backflip* and *Cartwheel*. Demonstration Replay (Random) randomly replaces a state in the rollout of the policy with the demonstration state at the same timestamp similar to the teacher forcing (Williams & Zipser, 1989). Demonstration Replay (Threshold) inserts the demonstration state based on the pose error. If the pose error between the demonstration state and the simulated state exceeds a threshold $\epsilon$, the simulated state will be replaced by the demonstration state. The Full Horizon Gradient variant backpropagates the gradient through the full horizon without any additional operation. We show the quantitative result in Fig. 7, and the qualitative result in Fig. 6.

**Demonstration replay helps stabilize policy learning and leads to better performance.** Compared with the Full Horizon Gradient without demonstration replay, the learning curve with demonstration replay is much smoother and finally converges to a lower pose error as shown in Fig. 7 (c)-(d). We show in Fig. 6 (b), this vanilla version of DiffMimic can easily get stuck into a local minimum. Instead of learning to backflip, the policy learns to bend down and use arms to support itself instead of jumping up. The other two variants, by contrast, both learn to jump and backflip in mid-air successfully.

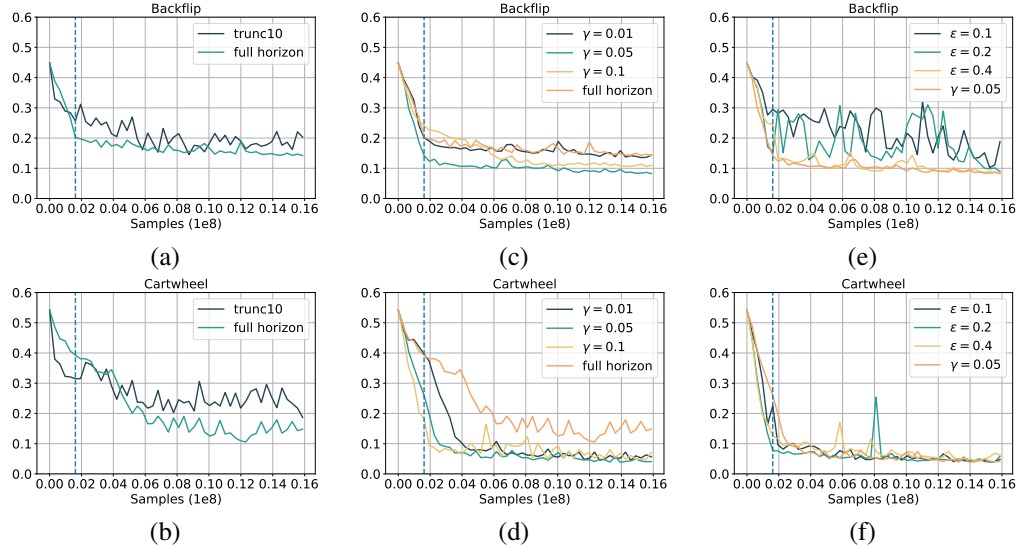

Figure 7: (a)-(b) Comparison between Full Horizon Gradient and truncation length of 10. (c)-(d) Comparison between Demonstration Replay (Random) and Full Horizon Gradient. (e)-(f) Comparison between Demonstration Replay (Random) and Demonstration Replay (Threshold). The blue dotted line denotes 10 minutes in the corresponding wallclock time.

**Policy-aware demonstration replay leads to a more faithful recovery of demonstration.** We compare Demonstration Replay (Threshold) with Demonstration Replay (Random) in Fig. 7 (e)-(f). Quantitatively, both variants yield similar results if the hyperparameter is properly tuned. However, the behavior of policies trained with these two strategies can be significantly different. In Fig. 6 (c) and (d), though both policies learn to backflip successfully, the policy trained with Demonstration Replay (Random) fails to recover the motion frame by frame. We show in Fig. 8 the per-frame pose error. Even though the average error over the whole trajectory for both variants is similar, Demonstration Replay (Threshold) gives a lower maximum per-frame error, which indicates a faithful recovery of the demonstration. This implies that simply minimizing the pose error may not suffice to learn a policy that tracks the demonstration closely. Finer-grained guidance based on the current performance of the policy is required.

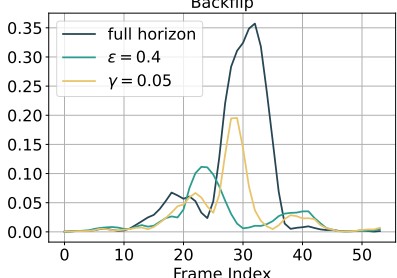

Figure 8: L2 error per frame in the rollout of the final policy. Although Demonstration Replay (Random) reduces the overall pose error compared to full-horizon optimization, the per-frame loss remains large in certain time steps due to the lack of a constraint. Demonstration Replay (Threshold) alleviates the issue.

## 5 CONCLUSION & FUTURE WORKS

In summary, we present DiffMimic utilizes differentiable physics simulators (DPS) for motion mimicking and outperforms several commonly used RL-based methods in various tasks with high accuracy, stability, and efficiency. We believe DiffMimic can serve as a starting point for motion mimicking with DPS and that our simulation environments provide exciting opportunities for future research.

While there are many exciting directions to explore, there are also various challenges. One such challenge is to enable motion mimicking within minutes given any arbitrary demonstration. With differentiable physics, this would greatly accelerate downstream tasks. In our work, we demonstrate that DiffMimic can learn challenging motions in just 10 minutes. However, the tasks we evaluated were relatively short and did not involve any interactions with other objects. As the dynamic system becomes more complex with multiple objects, we leave these challenges for future work.

## ETHICS STATEMENTS

We have carefully reviewed and adhered to the ICLR Code of Ethics. DiffMimic conducts experiments exclusively on humanoid characters, and no experiments involve human subjects. Additionally, all experiments are performed using open-sourced materials and are properly cited.

## REPRODUCIBILITY STATEMENTS

Our code is easily accessible and openly available for review at https://github.com/diffmimic/diffmimic. Additionally, the reported results can be easily reproduced using the provided code.

## ACKNOWLEDGMENT

This research is supported by the National Research Foundation, Singapore under its AI Singapore Programme (AISG Award No: AISG2-PhD-2022-01-036T, AISG2-PhD-2021-08-015T and AISG2-PhD-2021-08-018), NTU NAP, MOE AcRF Tier 2 (T2EP20221-0012), and under the RIE2020 Industry Alignment Fund - Industry Collaboration Projects (IAF-ICP) Funding Initiative, as well as cash and in-kind contribution from the industry partner(s).

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

## CONTENTS

## A  QUALITATIVE RESULTS

We present our qualitative results in the following website: https://diffmimic-demo-main-g7h0i8.streamlitapp.com/.

It contains the following qualitative results:

- Back-Flip 20-second rollout.
- Cartwheel 20-second rollout.
- Crawl 20-second rollout.
- Dance 20-second rollout.
- Jog 20-second rollout.
- Jump 20-second rollout.
- Roll 20-second rollout.
- Side-Flip 20-second rollout.
- Spin-Kick 20-second rollout.
- Walk 20-second rollout.
- Zombie 20-second rollout.
- Back-Flip 20-second rollout, trained for 2.5 hours.
- Back-Flip single cycle rollout, trained for 10 minutes.

## B    POLICY NETWORK

We use an MLP for the policy network. The network has two hidden layers with sizes 512 and 256. We use Swish (Ramachandran et al., 2017) for the activation function.

## C    HYPER-PARAMETERS

We use Adam optimizer (Kingma & Ba, 2014) in training, with a learning rate of 3e-4 that linearly decreases with training iterations. We apply gradient clipping and set the maximum gradient norm to 0.3. We set the batch size to the same as the number of parallel environments.

**Cyclic motions.** We set the maximum iterations to 5000. The number of environments is set to 200.

**Acyclic motions.** We set the maximum iterations to 1000. The number of environments is set to 300.

**Expert Replay.** We run experiments with two error thresholds, 0.2 and 0.4, and report the better of the two.

## D    STATE AND ACTION SPACE

**Humanoid.** We use 6d rotation representation (Zhou et al., 2019) in the state feature. The humanoid has an action size of 28 and a state feature size of 193.

**Humanoid with Sword and Shield.** We set up the Sword and Shield character following ASE (Peng et al., 2022). Compared to the default humanoid, the new character has 3 additional DOF on the right hand. A sword is attached to its right hand and a shield is attached to its left lower arm. The character has an action size of 31. The state feature size is 208.

# E   LEARNING CURVES

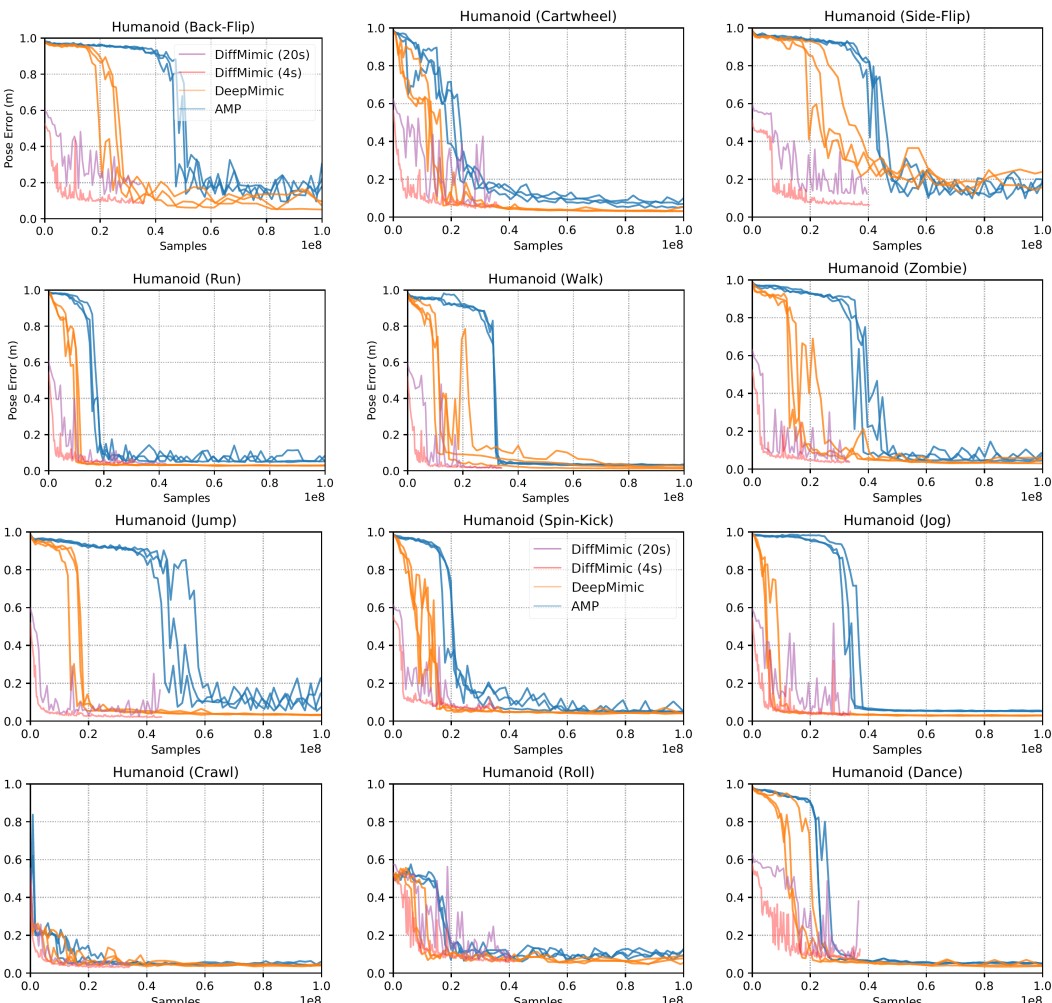

Figure 9: Pose error versus the number of samples. DiffMimic (4s): rollout 4 seconds of Diffmimic for evaluation. DiffMimic (20s): rollout 20 seconds of Diffmimic for evaluation.

## F    NEW CHARACTER

We include Ant as a new character. We use the trajectory in ILD (Chen et al., 2022) as the reference motion, which is obtained from a PPO (Schulman et al., 2017) agent that learns to move forward as fast as possible. We achieve a 0.059 meter final pose error. We show the training curve in Figure 10. and the qualitative result on the demo website under the tab "New Character".

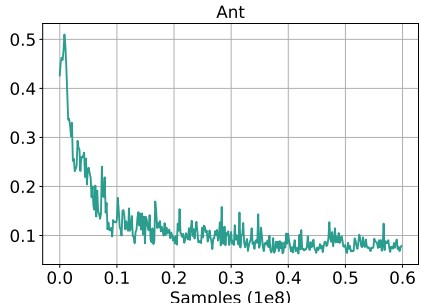

Figure 10: Pose error versus the number of samples for the new character, Ant.

## G    NEW SKILL

We run DiffMimic on a new skill, *360-degree Jump*, from the CMU Mocap Dataset (motion sequence CMU_075_09). The skill has been reported to be challenging for a RL-based method CoMic (Hasenclever et al., 2020) in a recent dataset paper (Wagener et al., 2022). DiffMimic is able to spin in the air and achieve a 0.020 meter final pose error. We show the training curve in Figure 11 and the qualitative result on the demo website under the tab "New Skill".

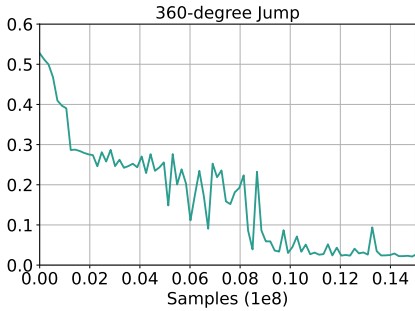

Figure 11: Pose error versus the number of samples for the new skill, *360-degree Jump*.

## H    NEW TRAINING STRATEGY

We additionally include the Reference State Initialization (RSI) technique (Peng et al., 2018a) into the DiffMimic training. RSI starts the training rollout from a random reference frame instead of the first reference frame. We observe a substantial empirical improvement after employing the new training strategy. The samples required are consistently around 90% less than DeepMimic (Peng et al., 2018a) as shown in Tab. 3. The final pose error has also been slightly improved as shown in Tab. 4. We show the qualitative results on the demo website under the tab "New Training". Notably, using RSI alone or RSI+Early Termination (ET) (Peng et al., 2018a) does not converge as shown in Figure 12.

Table 3: Number of samples required to roll out 20 seconds without falling in $(10^6)$ with Reference State Initialization (RSI) (Peng et al., 2018a). Percentage: change in the fraction of the DeepMimic samples.

| Motion | $T_{cycle}(s)$ | DeepMimic | Spacetime Bound | Ours | Ours w/ RSI |
|---|---|---|---|---|---|
| Back-Flip | 1.75 | 31.18 | 41.20 +32.1% | 14.88 -52.2% | 3.82 -87.7% |
| Cartwheel | 2.72 | 30.45 | 17.35 -43.0% | 13.92 -54.2% | 4.72 -84.5% |
| Walk | 1.25 | 23.80 | 4.08 -79.5% | 7.92 -66.7% | 1.55 -93.5% |
| Run | 0.80 | 19.31 | 4.11 -78.7% | 8.16 -57.7% | 1.41 -92.7% |
| Jump | 1.77 | 25.65 | 41.63 +77.8% | 5.28 -79.4% | 2.12 -91.7% |
| Dance | 1.62 | 24.59 | 10.00 -59.3% | 16.56 -32.6% | 2.19 -91.1% |

Table 4: Pose error comparison in meters. $T_{cycle}$ is the length of the reference motion for a single cycle. The error is averaged on 32 rollout episodes with a maximum length of 20 seconds. The result of Ours+RSI is obtained without DTW.

| Motion | $T_{cycle}(s)$ | DeepMimic | AMP | Ours | Ours w/o DTW | Ours w/ RSI |
|---|---|---|---|---|---|---|
| Back-Flip | 1.75 | $0.076 \pm 0.021$ | $0.150 \pm 0.028$ | $0.097 \pm 0.001$ | $0.105 \pm 0.022$ | $0.058 \pm 0.015$ |
| Cartwheel | 2.72 | $0.039 \pm 0.011$ | $0.067 \pm 0.014$ | $0.040 \pm 0.007$ | $0.040 \pm 0.007$ | $0.027 \pm 0.002$ |
| Crawl | 2.93 | $0.044 \pm 0.001$ | $0.049 \pm 0.007$ | $0.037 \pm 0.001$ | $0.037 \pm 0.001$ | $0.035 \pm 0.003$ |
| Dance | 1.62 | $0.038 \pm 0.001$ | $0.055 \pm 0.015$ | $0.070 \pm 0.003$ | $0.072 \pm 0.014$ | $0.042 \pm 0.001$ |
| Jog | 0.83 | $0.029 \pm 0.001$ | $0.056 \pm 0.001$ | $0.031 \pm 0.002$ | $0.031 \pm 0.002$ | $0.012 \pm 0.000$ |
| Jump | 1.77 | $0.033 \pm 0.001$ | $0.083 \pm 0.022$ | $0.025 \pm 0.000$ | $0.031 \pm 0.002$ | $0.015 \pm 0.000$ |
| Roll | 2.02 | $0.072 \pm 0.018$ | $0.088 \pm 0.008$ | $0.061 \pm 0.007$ | $0.083 \pm 0.001$ | $0.046 \pm 0.005$ |
| Run | 0.80 | $0.028 \pm 0.002$ | $0.075 \pm 0.015$ | $0.039 \pm 0.000$ | $0.046 \pm 0.000$ | $0.019 \pm 0.000$ |
| Side-Flip | 2.44 | $0.191 \pm 0.043$ | $0.124 \pm 0.012$ | $0.069 \pm 0.001$ | $0.121 \pm 0.009$ | $0.035 \pm 0.001$ |
| Spin-Kick | 1.28 | $0.042 \pm 0.001$ | $0.058 \pm 0.012$ | $0.056 \pm 0.000$ | $0.056 \pm 0.000$ | $0.036 \pm 0.000$ |
| Walk | 1.30 | $0.018 \pm 0.005$ | $0.030 \pm 0.001$ | $0.017 \pm 0.000$ | $0.017 \pm 0.000$ | $0.012 \pm 0.000$ |
| Zombie | 1.68 | $0.049 \pm 0.013$ | $0.058 \pm 0.014$ | $0.037 \pm 0.002$ | $0.037 \pm 0.002$ | $0.033 \pm 0.009$ |

## I  ANALYSIS ON ROBUSTNESS

We further evaluate DiffMimic's robustness to external forces. A force is applied to the pelvis of the character halfway through a motion cycle for 0.2 seconds, and we measure the maximum forwards and sideways push each policy can tolerate before falling. DiffMimic achieves comparable robustness to DeepMimic. We show the results in Table 5 and the qualitative results on the demo website under the tab "Robustness".

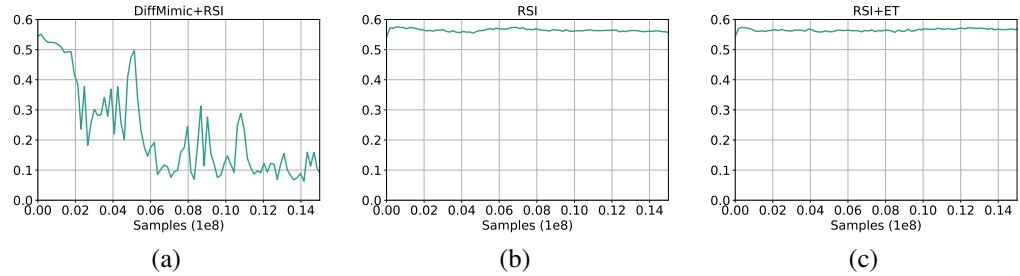

Figure 12: Pose error versus the number of samples on *Back-Flip* using (a) DiffMimic+RSI (b) standalone RSI (c) RSI+ET.

Table 5: Maximum forwards and sideways push each policy can tolerate before falling. Each push is applied halfway through a motion cycle to the character's pelvis for 0.2s. F: Forwards push. S: Sideways push. The unit is in Newton.

| Motion | DeepMimic (F) | Ours (F) | DeepMimic (S) | Ours (S) |
|---|---|---|---|---|
| Back-Flip | 440 | **750** | 100 | **310** |
| Cartwheel | 200 | **350** | 470 | **690** |
| Run | **720** | 500 | **300** | 270 |
| Spin-Kick | **690** | 630 | 600 | **730** |
| Walk | 240 | **250** | **300** | 220 |

## J ANALYSIS ON SENSITIVITY

DiffMimic with inaccurate estimates of physical parameters. Briefly speaking, DiffMimic is able to learn the control policy well even though the friction coefficient deviates from the original parameters. We show the results in Table 6 and the qualitative results on the demo website under the tab "Sensitivity".

Table 6: Sensitivity to the friction coefficient $\mu$, a system parameter. The evaluation is conducted on Zombie Walk, a motion heavily impacted by the friction coefficient. $\mu = 1.0$ is the standard setting, we additionally evaluate when $\mu = 0.8$ and $\mu = 1.2$.

| Friction | Pose Error (m) |
|---|---|
| $\mu$=0.8 | $0.047\pm 0.009$ |
| $\mu$=1.0 | $0.032\pm 0.009$ |
| $\mu$=1.2 | $0.028\pm 0.007$ |

## K ANALYSIS ON MOTION QUALITY

Motion quality can be severely affected by a small number of large error poses, which can not be reflected by the existing average pose error metric. We propose a Pose Absurdity metric to quantify large-error poses. L2@0.01, 0.05, and 0.1 measure the average L2 pose error on the worst 1%, 5%, and 10% frames respectively. Full Horizon Gradient, Demonstration Replay (Random) ratio $\lambda$, and Demonstration Replay (Threshold) threshold $\gamma$ are studied. Demonstration Replay (Threshold) outperforms other baselines by a large margin. We show the results in Table 7

## L SWORDING CHARACTER

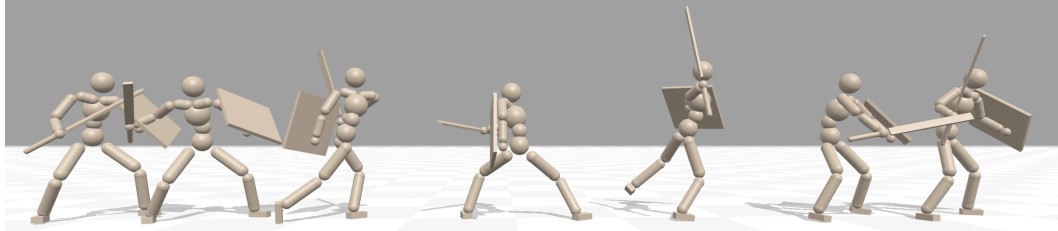

Figure 13: Character performing attack move with a sword and a shield. The reference motion clip is from Reallusion (Reallusion, 2022)

Table 7: Ablation experiment on Full Horizon Gradient, Demonstration Replay (Random) ratio $\lambda$, and Demonstration Replay (Threshold) threshold $\gamma$ with the new Pose Absurdity metric on *Back-Flip*. L2@0.01, 0.05 and 0.1 measure the average L2 pose error on the worst 1%, 5% and 10% frames respectively. Demonstration Replay (Threshold) significantly improves on the Pose Absurdity metric.

| Method | L2@0.01 | L2@0.05 | L2@0.1 |
|---|---|---|---|
| full horizon | 0.357 | 0.343 | 0.323 |
| $\lambda = 0.01$ | 0.379 | 0.341 | 0.318 |
| $\lambda = 0.05$ | 0.195 | 0.177 | 0.135 |
| $\lambda = 0.10$ | 0.359 | 0.332 | 0.262 |
| $\gamma = 0.1$ | 0.111 | 0.107 | 0.095 |
| $\gamma = 0.2$ | 0.113 | 0.107 | **0.093** |
| $\gamma = 0.4$ | **0.110** | **0.103** | 0.095 |

