# OpenReview forum: "DiffMimic: Efficient Motion Mimicking with Differentiable Physics"
_ICLR.cc/2023/Conference — ICLR 2023 poster_

### Official Review · Reviewer_8KfC · 2022-10-22

**Confidence:** 4
**Correctness:** 3
**Technical Novelty And Significance:** 2
**Empirical Novelty And Significance:** 2
**Recommendation:** 5

**Clarity, Quality, Novelty And Reproducibility:**

- Clarity & Quality: The paper is generally clear and easy to follow. The accompanying visualization through the interactive website is very helpful.
- Novelty: As I said above, the paper lacks novelty since prior work [1, 2] has already explored applying a differentiable physics simulator to model human motions in much more complex settings. Technical innovation is also incremental.
- Reproducibility: I believe the paper can be reproduced relatively easily for people familiar with motion and control.

**Strength And Weaknesses:**

**Strength:**

- The method achieves faster convergence by using a differentiable physics simulator compared to RL-based approaches.
- The paper is generally well-written and easy to follow.

**Weakness:**

- The novelty of the paper is limited. The idea of using a differentiable physics simulator to model human motions has been explored in prior work [1, 2] and in more complex settings, human pose estimation, navigation, path following, etc. There is no new method proposed in this paper besides using a differentiable physics simulator. The demonstration replay seems like an incremental technical contribution, and its benefit is not very significant given the mean pose error stays the same thus traditional approaches perform similarly over the whole motion.
- From Table 1, we can see that DeepMimic actually outperforms the proposed method in at least 6 motions (Back-Flip, Dance, Run, etc.) where in some cases DeepMimic actually beats the method a lot (0.038 vs 0.070 for Dance), so this method overall has only slightly better performance than an RL-based approach like DeepMimic. Even though the proposed method learns faster due to the use of simulation, achieving better motion quality consistently in the end is more important for motion imitation.
- From the visualization on the website, the “jog” motion produced by this method seems to be rather unnatural with the character limping, this does not happen for DeepMimic or AMP. There is also some foot sliding in the “roll” motion. Are the artifacts caused by the use of a differentiable physics simulator, which might not enforce the contact constraints as well?

[1] Gärtner, Erik, et al. "Differentiable Dynamics for Articulated 3d Human Motion Reconstruction." CVPR. 2022.

[2] Won, Jungdam, Deepak Gopinath, and Jessica Hodgins. "Physics-based character controllers using conditional VAEs." SIGGRAPH 2022.

**Summary Of The Paper:**

This paper proposes a method for learning human locomotion policies from human motion data using an off-the-shelf differentiable physics simulator (DPS). Specifically, the method uses the Brax simulator (Freeman et al., 2021) to enable gradient computation of the forward dynamics. A simple joint and angle-based motion matching loss is used to supervise the training. To further improve the stability of training, the paper proposes a technique called demonstration replay, which resets the rollout state to the reference state when it deviates too much. Experiments on human locomotions and acrobatics show the method converges faster than RL-based methods while achieving similar motion quality.

**Summary Of The Review:**

The paper shows improved efficiency of motion imitation with a differentiable physics simulator. However, the motion quality improvement is not huge, and the paper didn’t enable learning new motions that RL-based methods can’t. A main concern is also the paper’s lack of novelty. Therefore, I lean towards reject at this point.

--- update ---
After the rebuttal, I'm slightly more positive about this paper. I don't mind seeing the paper accepted. I still think the motion quality needs to be improved (e.g., obvious foot sliding). It is possible the lower quality is due to DPS's approximation of contact to enable differentiability, and it could be an inherent limitation of using DPS.

---

> ### Author Response · Authors · 2022-11-17
> **Author Response (Part 2/2)**
>
> > **Q4**: Are the artifacts caused by the use of a differentiable physics simulator, which might not enforce the contact constraints as well?
>
> **A4**: It is possible that differentiable physics simulators like Brax have different contact results from Bullet used in DeepMimic and AMP, since they have different rigid body contact model implementations. Nonetheless, DiffMimic leverages the differentiability and does not depend on specific contact models. The "artifacts" assumingly caused by DPS can be solved if future DPS could develop more accurate contact computation.
>
>
> > **Q5**:  the paper didn’t enable learning new motions that RL-based methods can’t.
>
> **A5**: RL-based methods are applicable in learning many motions. However, they are generally much more expensive to learn new motions. For example, it would require heavy reward engineering or notoriously data-hungry. Thus, we introduce DiffMimic to significantly simplify the policy training pipeline, increase the sample efficiency and alleviate the heavy reward engineering.
>
> In addition, we do observe some cases in which our method is able to learn motions that cannot be learned by the RL-based method. However, we decided not to put it in the main text to avoid distracting the main focus of the paper. The motion is from [1] and cannot be learned by CoMic (Hasenclever et al., 2020) as stated in the paper. We refer the reviewer to the link (https://diffmimic-demo-main-g7h0i8.streamlit.app/) for the visualization.
>
> [1] Nolan Wagener, Andrey Kolobov, Felipe Vieira Frujeri, Ricky Loynd, Ching-An Cheng, Matthew Hausknecht, MoCapAct: A Multi-Task Dataset for Simulated Humanoid Control NeurIPS, 2022.

---

> > ### Comment · Reviewer_8KfC · 2022-11-20
> > **Response**
> >
> > I thank the authors for the rebuttal, which addressed some of my concerns. Hence, I increased my score. I still think the motion quality of the learned motions needs to be improved, there are quite some foot sliding in the generated motions (e.g., 360-degree jump), which possibly comes from the approximation of contact. DPS makes these approximations to enable differentiability, so it could be an inherent limitation of using DPS.

---

> > > ### Author Response · Authors · 2022-11-22
> > > **Glad to address some of your concerns!**
> > >
> > > We are glad that some of your concerns have been addressed and thank you for raising the score! Improving the contact approximation in DPS is indeed still under active research. We would like to reiterate that DiffMimic is a general solution for motion mimicking that leverages the differentiability of DPS and its benefits are independent of the specific DPS we use. Given improved DPS contact models, we would expect improved generated motions with DiffMimic.

---

> ### Author Response · Authors · 2022-11-17
> **Author Response (Part 1/2)**
>
> We thank the reviewer for the valuable comments! We detail the response in the following threads. We kindly ask the reviewer to let us know if further clarification or information is needed.
>
> > **Q1**: The novelty of the paper is limited. The idea of using a differentiable physics simulator to model human motions has been explored in prior work [1, 2] and in more complex settings, human pose estimation, navigation, path following, etc.
>
> **A1**: Thank you for bringing these papers to our attention. However, they are not directly comparable. In paper [1], the DPS is only used to optimize the estimated control trajectory. It does not enable policy learning. We show that the learned policy is robust to an external force (https://diffmimic-demo-main-g7h0i8.streamlit.app/). Simply optimizing the control trajectory is unable to recover from such external force during the evaluation stage. In addition, utilizing the existing DPS for policy learning is non-trivial considering the gradient explosion/vanishing and local optimal problem. Therefore, we introduce the adaptive expert demo replay. Without the adaptive expert demo replay mechanism, the sample efficiency will be much lower and the policy may not even converge.
>
> Paper [2], as described in the related work section, does not utilize a differentiable physics simulator but rather learns a world model. It first uses the DeepMimic-style RL method to learn the control policy for each motion caption data (Sect 5.1.2). Then it collects a state-action database by rolling out each policy multiple times and trains a differentiable physics simulation layer on the database (Sect 5.1.2). As discussed in related works, collecting a large state-action database can be expensive in practice. For example, a recent state-action database MoCapAct requires about 50 years of wallclock time. Moreover, the learned world model is prone to error accumulation, as pointed out in SuperTrack. In the case of [2], the world model is only used for single-step prediction, which might not be sufficient to mimic challenging motions that require long-term apprehension.
>
> > **Q2**: The demonstration replay seems like an incremental technical contribution, and its benefit is not very significant given the mean pose error stays the same thus traditional approaches perform similarly over the whole motion.
>
> **A2**: This is a great point for us to elaborate more on! By observing the average error, the improvement seems incremental. However, this is due to the metric we used. The mean joint position error can only reflect the average error but the visual quality usually depends on the ‘absurd’ poses, i.e., outlier poses with large errors. The baseline methods suffer from absurd poses a lot as shown in Figure 6. A per-frame loss visualization in Figure 8. reveals our proposed Demonstration Replay (Threshold) significantly reduces the number of large-error poses.
>
> Here, we further propose a Pose Absurdity metric to quantify large-error poses. L2@0.01, 0.05, and 0.1 measure the average L2 pose error on the worst 1%, 5%, and 10% frames
> respectively. Full Horizon Gradient, Demonstration Replay (Random) ratio $\lambda$, and Demonstration Replay (Threshold) threshold $\gamma$ are studied. Demonstration Replay (Threshold) outperforms other baselines by a large margin. We also refer the reviewer to **Table 7** in the appendix.
>
> | Backflip     | L2@0.01 | L2@0.05 | L2@0.10 |   |
> |--------------|---------|---------|---------|---|
> | full horizon | 0.357   | 0.343   | 0.323   |   |
> | λ = 0.01     | 0.379   | 0.341   | 0.318   |   |
> | λ = 0.05     | 0.195   | 0.177   | 0.135   |   |
> | λ = 0.1      | 0.359   | 0.332   | 0.262   |   |
> | ɣ = 0.1      | 0.111   | 0.107   | 0.095   |   |
> | ɣ = 0.2      | 0.113   | 0.107   | 0.093   |   |
> | ɣ = 0.4      | 0.110   | 0.103   | 0.095   |   |
>
>
> > **Q3**:  This method overall has only slightly better performance than an RL-based approach like DeepMimic.
>
> **A3**: We would like to clarify the main contribution of DiffMimic is to simplify the policy training pipeline and increase the sample efficiency for motion mimicking. DiffMimic is much more sample efficient compared with the previous RL-based approach. Sample inefficiency is an important limitation for the RL-based approach to be widely applied.  For example, it would only require 14.88 million samples while DeepMimic would require 31.18 million samples to learn ‘backflip’ if both use Reference State Initialization. As we have demonstrated in the visualized results, DiffMimic is able to learn backflips within 10 minutes of training on one single GPU.
>
> In addition, by incorporating the RSI training technique, the sample required to perform 20s 'backflip' further reduces to 3.82 million and DiffMimic outperforms DeepMimic in terms of pose error over 11 out of 12 different motions.

---

### Official Review · Reviewer_a2aF · 2022-10-23

**Confidence:** 3
**Correctness:** 3
**Technical Novelty And Significance:** 3
**Empirical Novelty And Significance:** 3
**Recommendation:** 8

**Clarity, Quality, Novelty And Reproducibility:**

Clarity

The idea and methodology of this paper seem to be intuitive and clear to me.

Quality

The authors can improve the quality of this paper during writing. For example, there is one missing citation in Section 2, paragraph 1, line 7. Moreover, curve plots (Figures 4 and 7) look kind of messy, and hard to distinguish the difference between methods. It might help if the authors could run multiple experiments with different random seeds to smooth the curve and draw the standard deviation with a shaded area.

Novelty

The demonstration reply technique is simple but effective. Using DPS for motion mimicking is interesting. There is a concurrent work, ILD, as mentioned. It would be better, but probably not necessary since, to compare with it.

Reproducibility

Given that the authors promise to release the code and the method is not too complicated, I think the reproducibility should be fine.


**Strength And Weaknesses:**

Strengths

I think the biggest strength of this paper is that it presents a new direction toward motion mimicking problems. Compared to the previous reinforcement learning methods, the gradients from DPS can provide more information to the learning process and thus improve the sample efficiency.

Another strength is that the proposed approach can achieve reasonably good results (as shown in the supplementary video) for rather complex and dynamic motions. This method can also use constantly lower sampling numbers during the training than Deepmimic.

Weaknesses

One of my concerns is the generalizability of this method. Since the dynamics and implementation details of the DPS might not be exactly the same as the original demonstration, it is not fully investigated how would the learned policy perform in the original simulation environment.

Moreover, the authors mentioned relaxation in their DPS environment, “The joint limits of the character are relaxed to allow smoother gradient propagation.” I wonder whether the measurement is performed in the same environment as the other comparison methods. How much difference this modification would bring to the learning task?


**Summary Of The Paper:**

This paper proposes an algorithm for motion mimicking using differentiable physics simulators. With DPS and its gradients, the authors convert a policy learning problem to a much easier state matching problem. Instead of running an entire simulation during training, this paper proposes a Demonstration Replay mechanism to mitigate error accumulation. The Demonstration Replay replaces a simulation frame with the ground truth demonstration if its error is beyond a threshold.

The authors show that the proposed DiffMimic outperforms previous RL-based motion mimicking methods in terms of sample efficiency.


**Summary Of The Review:**

This paper provides a new way to solve motion mimicking problems using differentiable physics simulations. The method looks concise and intuitive. The results are also good. It could be better if the paper presentation can be improved and more comparisons can be added. In summary, I think this paper looks fine but can be made more solid.

---

> ### Author Response · Authors · 2022-11-17
> **Author Response**
>
> We thank the reviewer for the reflective comments! We will clean up the plots after all the experiments are finalized as more experiments are added. We detail the response in the following threads. We kindly ask the reviewer to let us know if further clarification or information is needed.
>
> > **Q1**: One of my concerns is the generalizability of this method. Since the dynamics and implementation details of the DPS might not be exactly the same as the original demonstration, it is not fully investigated how would the learned policy perform in the original simulation environment.
>
> **A1**: We completely agree that sim-to-real is of great importance to make the DiffMimic more useful. However, considering we mainly target animation applications, the sim-to-real problem is less demanding since the deployment environment would be highly similar to the training environment.
>
> Having said that, we envision that the sim-to-real problem can be mitigated with DPS. For example, the sim-to-real gap can be efficiently minimized through system identification [1,2] with DPS.  DiffMimic opens up the possibility to adjust the character's weight distribution/actuator force/damping factors along with mimicking. The parameter of the environment and the policy can be learned together to tackle the sim-to-real problem. However, methods for incorporating system identification into the learning loop can be highly task-dependent and may require a separate set of analyses beyond the scope of our contribution of DPS-based motion mimicking. We consider it an important future direction.
>
> [1] Murthy, J. Krishna, et al. "gradSim: Differentiable simulation for system identification and visuomotor control." International Conference on Learning Representations. 2020.
> [2] Le Lidec, Quentin, et al. "Differentiable simulation for physical system identification." IEEE Robotics and Automation Letters 6.2 (2021): 3413-3420.
>
>
> > **Q2**:  “The joint limits of the character are relaxed to allow smoother gradient propagation.” I wonder whether the measurement is performed in the same environment as the other comparison methods. How much difference this modification would bring to the learning task?
>
> **A2**: Joint limit is commonly ignored in humanoid characters since most of the joints are spherical joints and can not be well bounded. For example, DeepMimic only has four joint limits for knees and elbows (https://github.com/xbpeng/DeepMimic/issues/146#issuecomment-807446006). In this work, we ignore the four as well. In particular, the joint limit has a trivial effect on motion mimicking task since the joint limit will be fulfilled once the motion is mimicked (https://github.com/bulletphysics/bullet3/issues/2303#issuecomment-564044777).  After adding the four joint limits during inference, we do not observe any visible differences in the rollout.
>
> We would also like to refer the reviewer to the appendix of the updated manuscript with more comparison and analysis including:
> 1. Experiment on a new character.
> 2. Experiment on a new skill, 360-degree Jump. We reveal that DiffMimic is able to mimic some challenging motions that RL-based methods fail to.
> 3. Experiment on a new training strategy, DiffMimic with RSI, which requires 90% fewer samples than DeepMimic and has better accuracy. We show that DiffMimic is a promising direction and has great potential when combined with better training recipes.
> 4. Analysis on system sensitivity. We show that DiffMimic performs consistently when system parameters like the friction coefficient are inaccurately estimated.
> 5. Analysis on perturbation robustness. We show that DiffMimic learns policies that are robust to external pushes during rollout.
> 6. Analysis on motion quality. We use a pose absurdity metric to quantify the motion quality improvement of Demonstration Replay.

---

> > ### Comment · Reviewer_a2aF · 2022-12-03
> > **Thanks for your response. Score raised.**
> >
> > Thanks for your detailed response and the additional experiments, which make the paper much stronger.

---

> > > ### Author Response · Authors · 2022-12-06
> > > **Thank you for raising the score!**
> > >
> > > Thank you for your recognition of our response and additional experiments!

---

### Official Review · Reviewer_s4rT · 2022-10-24

**Confidence:** 4
**Correctness:** 3
**Technical Novelty And Significance:** 2
**Empirical Novelty And Significance:** 3
**Recommendation:** 8

**Clarity, Quality, Novelty And Reproducibility:**

The paper is reasonable well written. Some minor english fixes need to be made.
The code is available on GitHub.  Further details might nevertheless be useful, so that readers don't need to go to the code to understand particular hyperparameters,

**Strength And Weaknesses:**

Strengths:
- speeds up learning for an important class of problem (physics-based motion imitation)
- multiple useful practical insights on window length and resets, i.e., (demonstration replay) that are needed
  to get DPS-based methods working with complex human motions. This is the first time that I have
  seen Brax successfully used for this kind of scenario, although [Xu et al 2022] demonstrate results
  of similar complexity with a simulator of their own.

Weaknesses:
- mentions of closely related work, but with no deeper discussion on comparing the approaches,
  or direct experimental comparisons
  [Fussell et al 2021, Xu et al 2022, Mora et al 2021]
  as well as "A Scalable Approach to Control Diverse Behaviors for Physically Simulated Characters"
  There are differences of the current work with respect to all of these, but much of the value of the paper
  lies in discussing these differences. I.e., Xu et al 2022 come up with a different solution to the
  basic challenges, i.e., exploding gradients, local minima.


**Summary Of The Paper:**

A differentiable physics simulator (DPS, Brax in this case), is used to learn an RL policy that
imitates a given reference motion, as done in DeepMimic. The contributions include (a) better sample
efficiency, and therefore faster learning, as compared to DeepMimic; (b) the use of "demonstration
replay" when the motion deviates too far from the reference motions.  This is a type of reset that
is critical to provide stable gradients and to avoid local minima.


**Summary Of The Review:**

I see the value of this paper in terms of the pragmatic details that are needed to overcome the basic issues of exploiting differentiable simulators to directly learn policies. Getting these details right is important, as we also see from [Xu et al 2022].  However, the discussion could really be improved.

The paper claims that DeepMimic rewards are heavily engineered. I believe that the
DeepMimic imitation reward is quite similar to the loss function used here.

DeepMimic introduces both early termination (ET) and reference state initialization (RSI)
precisely to escape local minima of the type described here.  The idea of using a deviation threshold
to dynamically reset back to the trajectory could also be seen as a combination of ET and RSI.

Section 2, first para, missing bibtex reference e.g., "?]"

The existing work that makes use of differentiable simulators is mentioned, but then more-or-less ignored.
I'm not clear why. E.g.,   [Fussell et al 2021, Xu et al 2022, Mora et al 2021].
The following work also takes advantage of transfer learning when learning to imitate a wide range of motions,
and to therefore to immediately generalize to a large percentage of new motions:
"A Scalable Approach to Control Diverse Behaviors for Physically Simulated Characters"
In the long term, this may be more useful than speeding up the single-motion case.

Fig 4 shows 3 runs for some methods and only one run for DiffMimic.
DiffMimic presumably not deterministic, i.e., it still uses a stochastic action policy,
to help enable encourage robustness. Or is that incorrect?

text below Figure 4:
"Dynamic Time Warping is applied to sync the simulated motion and the reference motion following the convention"
Which convention?  I don't believe that DeepMimic uses DTW.

Table 2:
The results might be better stated as a fraction of the DeepMimic time.

How does the robustness of the learned policies compare to those learned with DeepMimic?
Is the exploration noise comparable?

---

> ### Author Response · Authors · 2022-11-17
> **Author Response (Part 2/2)**
>
> > **Q4**: text below Figure 4: "Dynamic Time Warping is applied to sync the simulated motion and the reference motion following the convention" Which convention? I don't believe that DeepMimic uses DTW.
>
> **A4**: The results are taken from [1] (Table 3) when the mimicking results from DeepMimic and AMP are compared. "Performance is recorded as the average pose error (in units of meters) between the time-warped trajectories from the reference motion and simulated character."
>
> In fact, DTW has a negligible performance boost to the results of DiffMimic since it is time synchronized, we add it to keep the comparison consistent. We have updated the manuscript to report the result of the DiffMimic without DTW in the appendix. We refer the reviewer to Table 4 in the appendix for details.
>
> [1] Xue Bin Peng, Ze Ma, Pieter Abbeel, Sergey Levine, Angjoo Kanazawa, AMP: Adversarial Motion Priors for Stylized Physics-Based Character Control, SIGGRAPH, 2021
>
>
>
> > **Q5**: DiffMimic presumably not deterministic, i.e., it still uses a stochastic action policy, to help enable encourage robustness？How does the robustness of the learned policies compare to those learned with DeepMimic? Is the exploration noise comparable?
>
> **A5**: Thank you for raising the question. The robustness of the policy is an important aspect. DiffMimic does employ a stochastic policy to encourage robustness. The variance in 32 rollouts of the same policy can be found in Table 1. Our website demo also provides different rollouts from different random seeds.
>
> We further evaluate DiffMimic's robustness to external forces. A force is applied to the pelvis of the character halfway through a motion cycle for 0.2 seconds, and we measure the maximum forwards and sideways push each policy can tolerate before falling. The results are presented in the appendix. DiffMimic achieves comparable robustness to DeepMimic.  We refer the reviewer to **Table 5** in the appendix and the visualization (https://diffmimic-demo-main-g7h0i8.streamlit.app/).

---

> > ### Comment · Reviewer_s4rT · 2022-12-03
> > **Thanks for detailed response; updated score**
> >
> > The detailed response is appreciated. I now score the paper a 7, but that doesn't exist on the scale, so I have updated my score to 8.  The value of the paper is enhanced with the more detailed comparative description of the related work, as given in the new Appendix M, so do include that.
> >
> > One last question that would be useful to think about, for the field.  Are the DeepMimic and DiffMimic policies transferable across simulators? Brax, Bullet, and MuJoCo all make different decisions with respect to handling contacts. Thus to what extent does the robustness of the learned policies extend to "working across simulators"? Some of the differences in learning rates might be attributed to the simulator differences.

---

> > > ### Author Response · Authors · 2022-12-06
> > > **Thank you for raising the score!**
> > >
> > > Thank you for your recognition of our response! We will include Appendix M in the main paper in the final version.
> > >
> > > We agree that transferability between different simulators is an important problem to be addressed in practice and can be useful in real-world applications. Theoretically, DiffMimic is agnostic to the contact simulation method as long as the simulator is differentiable. However, as different contact simulation methods result in different levels of smoothness of the gradients, hyper-parameters, e.g., learning rate, gradient clipping rate, etc, might need to be tuned. We leave it for future study.

---

> ### Author Response · Authors · 2022-11-17
> **Author Response (Part 1/2)**
>
> We thank the reviewer for the thoughtful comments! We detail the response in the following. We kindly ask the reviewer to let us know if further clarification or information is needed.
>
> > **Q1**: Discussion on comparison with SHAC, PODS, SuperTrack, and ScaDiver.
>
> **A1**: We agree with the reviewer a deeper discussion with existing relevant approaches is important. We have added one paragraph to the manuscript in appendix **Section M** and we will merge it into the main text in the final version.
>
>
> > **Q2**: The paper claims that DeepMimic rewards are heavily engineered. I believe that the DeepMimic imitation reward is quite similar to the loss function used here.
>
> **A2**: The form of the loss function and the imitation reward is indeed similar. However, a proper weight scheme over different reward terms is required to guide the training in DeepMimic.
> To ensure different reward terms are in a similar numerical scale and do not overwhelm other terms, a coefficient should be manually selected when calculating each reward term. We refer the reviewer to **Section L** in the appendix for details.
>
> Getting these numbers right is non-trivial and highly task-dependent. It would play an important role to train a performant policy [1,2]. While in DiffMimic, we can simply sum the l2 distance of position and rotation as the loss term thanks to the analytical gradient provided by DPS.
>
> [1] Logan Engstrom, Andrew Ilyas, Shibani Santurkar, Dimitris Tsipras, Firdaus Janoos, Larry Rudolph, Aleksander Madry, Implementation Matters in Deep Policy Gradients: A Case Study on PPO and TRPO, ICLR, 2020.
> [2] Marcin Andrychowicz, Anton Raichuk, Piotr Stańczyk, Manu Orsini, Sertan Girgin, Raphael Marinier, Léonard Hussenot, Matthieu Geist, Olivier Pietquin, Marcin Michalski, Sylvain Gelly, Olivier Bachem,What Matters In On-Policy Reinforcement Learning? A Large-Scale Empirical Study, ICLR, 2021.
>
>
> > **Q3**:  DeepMimic introduces both early termination (ET) and reference state initialization (RSI) precisely to escape local minima of the type described here. The idea of using a deviation threshold to dynamically reset back to the trajectory could also be seen as a combination of ET and RSI.
>
> **A3**: RSI+ET specifies initialization and termination states, while Demonstration Replay is a Teacher Forcing style strategy that does not change the start and the end of an episode. More specifically, RSI+ET initiates from a random reference state and ends when the character falls, while our approach replays a reference state whenever the character's pose deviates too much from the reference.
>
> The differences root in different learning paradigms. RSI+ET is developed from the RL perspective where an agent learns to explore the states to maximize reward. Placing agents at random states at the beginning of an episode benefits exploration. Demonstration Replay is developed from a supervised learning perspective, where a model is trained to reconstruct a training sequence. Proper insertion of ground truth data stabilizes training. Empirically, we show that RSI+ET alone does not yield ideal results with DPS as shown in Figure 13 in the appendix, which indicates that the strategy is not sufficient to address the local optimal challenge in the supervised learning-based motion mimicking.
>
> Moreover, we show incorporating RSI into DiffMimic improves the performance on top of Diffmimic, which suggest that RSI is still an effective technique once the optimization challenge in DPS is solved by DiffMImic. We refer the reviewer to **Table 3 & 4** in the appendix for quantitative results.

---

### Official Review · Reviewer_JNig · 2022-10-24

**Confidence:** 4
**Correctness:** 3
**Technical Novelty And Significance:** 2
**Empirical Novelty And Significance:** 3
**Recommendation:** 6

**Clarity, Quality, Novelty And Reproducibility:**

- Some of the figures confuse me. For example, for Figure 7, I would recommend arranging the figures in columns instead of in rows for clarification.
- The current content and format of the figure look great. A minor suggestion is to show the loss in log scale so the readers can tell them apart.

**Strength And Weaknesses:**

Strength:
- This manuscript described a feasible pipeline for motion mimicking. From the perspective of differentiable, this is a good task and should be a correct practice of using differentiable physics.
- The proposed Demonstration Replay looks
- The authors presented extensive experiments with complex motions.

Weaknesses:
- My major reservation is about technical novelty. "Demonstration" is a developed method for differentiable physics and has shown the power of manipulation and locomotion tasks. The proposed demonstration replay, although effective on the shown tasks, has a marginal gain over the random baseline.
- Training a controller using differentiable physics can be considered model-based reinforcement learning with a perfect world model. I would expect a model-based RL baseline to demonstrate the claimed "better sample efficiency".
- I admit Humanoid is a complex robot to work on. It will be great to see the proposed method applied to more robots, e.g., ants or cheetahs.

**Summary Of The Paper:**

This manuscript proposed a method for motion mimicking using differentiable physics. This manuscript used Brax as the backbone differentiable physics simulator and proposed to use demonstration to enhance the training. This manuscript showed the comparison with several baselines including DeepMimic and AMP, which are well-known as the established methods in this field. These experiments showed better sample efficiency compared to the gradient-free methods. Additionally, this manuscript examined the proposed method by a set of ablation studies of truncation length and demonstration replay.

**Summary Of The Review:**

This paper proposed a sensible method for solving motion mimicking using differentiable physics. Their results are backed by extensive experiments on a complex robot. My main complain is on the technical novelty side. However, I think the results can contribute to the community.

---

> ### Author Response · Authors · 2022-11-17
> **Author Response**
>
> We thank the reviewer for the constructive comments! We detail the response in the following. We kindly ask the reviewer to let us know if further clarification or information is needed.
>
>
> > **Q1**: The proposed demonstration replay, although effective on the shown tasks, has a marginal gain over the random baseline.
>
> **A1**: This is a great point for us to clarify more! By observing the average error for both methods, the improvement seems incremental. However, this is due to the metric we used. The mean joint position error can only reflect the average error but the visual quality usually depends on the ‘absurd’ poses, i.e., outlier poses with large errors. The baseline methods suffer from absurd poses a lot as shown in Figure 6. A per-frame loss visualization in Figure 8. reveals our proposed Demonstration Replay (Threshold) significantly reduces the number of large-error poses.
>
> Here, we further propose a Pose Absurdity metric to quantify large-error poses. L2@0.01, 0.05, and 0.1 measure the average L2 pose error on the worst 1%, 5%, and 10% frames
> respectively. Full Horizon Gradient, Demonstration Replay (Random) ratio $\lambda$, and Demonstration Replay (Threshold) threshold $\gamma$ are studied. Demonstration Replay (Threshold) outperforms other baselines by a large margin. We also refer the reviewer to **Table 7** in the appendix.
>
> | Backflip     | L2@0.01 | L2@0.05 | L2@0.10 |   |
> |--------------|---------|---------|---------|---|
> | full horizon | 0.357   | 0.343   | 0.323   |   |
> | λ = 0.01     | 0.379   | 0.341   | 0.318   |   |
> | λ = 0.05     | 0.195   | 0.177   | 0.135   |   |
> | λ = 0.1      | 0.359   | 0.332   | 0.262   |   |
> | ɣ = 0.1      | 0.111   | 0.107   | 0.095   |   |
> | ɣ = 0.2      | 0.113   | 0.107   | 0.093   |   |
> | ɣ = 0.4      | 0.110   | 0.103   | 0.095   |   |
>
>
> > **Q2**:   I would expect a model-based RL baseline to demonstrate the claimed "better sample efficiency".
>
> **A2**: Compared with methods like Supertrack (Fussel et al., 2021) and Phys-CVAE (Won et al., 2022) which learns a world model to train the policy, DiffMimic does not require learning the world model but uses Differentiable simulations. Differentiable simulations utilize physical models to provide more reliable gradients with better interpretability compared with learned black-box dynamics. The learned model can suffer from accumulated errors during the training, as pointed out in the limitation section in SuperTrack. On the other hand, the analytical gradient from DPS suffers from noise or can sometimes be wrong in a contact-rich environment. Thus, utilizing the existing DPS is non-trivial and can be more challenging considering the gradient explosion/vanishing and local optimal problem.  Therefore, we introduce the adaptive expert demo replay. Without the adaptive expert demo replay mechanism, the sample efficiency will be much lower and the policy may not converge. Unfortunately, neither Supertrack nor Phys-CVAE releases their code, thus, we are unable to directly compare DiffMimic with them.  We refer the reviewer to **Figure 7** for the sample efficiency of the full horizon gradient implementation of DiffMimic for comparison.
>
>
> > **Q3**:  It will be great to see the proposed method applied to more robots, e.g., ants or cheetahs.
>
> **A3**: We focus on motion mimicking, which lacks an explicit reward definition and expert actions. Instead, it considers only expert poses and actions that cannot be easily decoded. As a result, standard continuous control tasks are not directly comparable.
>
> We add an additional motion mimicking experiment on ants and calculate the average pose error. We refer the reviewer to the appendix **Section F** for details and the link (https://diffmimic-demo-main-g7h0i8.streamlit.app/) for visualization.

---

> > ### Comment · Reviewer_JNig · 2022-12-06
> > **Thanks!**
> >
> > I have read the responses from the authors, and I think I am happy with the answers I have got. I appreciate the additional experiments from the authors. I would like to raise my score a little bit higher to 7, but there exists none. I will keep a 6 but lean towards acceptance.

---

> > > ### Author Response · Authors · 2022-12-06
> > > **Thank you for recognizing our response!**
> > >
> > > Thank you for recognizing our response! We really appreciate your detailed comments and suggestions for improvement. We will include the additional revisions in the final version.

---

### Official Review · Reviewer_yUu6 · 2022-10-25

**Confidence:** 4
**Correctness:** 4
**Technical Novelty And Significance:** 1
**Empirical Novelty And Significance:** 2
**Recommendation:** 6

**Clarity, Quality, Novelty And Reproducibility:**

The paper is well-presented. However, the novelty is limited because the idea is a little straightforward. I believe that adding adaptivity to the teacher-forcing mechanism should have been explored in the field of NLP.

One of the important strengths of RL-based methods is that they can be easily transferred onto real robots. However, the policy rollout on real robots is not usually differentiable, which limits the application of the proposed method. It would be more impressive if the authors could use differentiable physics to remove the assumption of known parameters of rigid body dynamics, which could shrink the sim-to-real transfer gap.

**Strength And Weaknesses:**

**Strengths:**

The evaluation of the proposed method is thorough. The authors evaluated their method on vast types of tasks and compared with several baselines.

**Weakness:**

- How is the data generated?  Is it generated by simulations or motion capture systems?
- How are the parameters of rigid body dynamics such as friction coefficients/restitution coefficients generated? I would also like to see whether the proposed method is sensitive to physical parameters, i.e., when the estimations of these parameters are not perfect, will the method still gives reasonable results?
- Please also see the evaluation of novelty below.

**Summary Of The Paper:**

This paper proposes a differentiable physics based framework to solve motion mimicking problems without reinforcement learning. The authors show that their method is much more efficient than reinforcement learning based methods. To solve the local minimum issue caused by the long horizon, the authors proposed an adaptive teacher-forcing mechanism called Demonstration Replay, which adaptively separates the whole trajectory into several sub-trajectories with the ground truth states as the input to aid the training.

**Summary Of The Review:**

This paper proposes a differentiable physics-based framework for motion mocking tasks. The results show that the diff-physics based method is more efficient than RL-based methods. However, replacing the state transitions in RL frameworks with differentiable physics seems pretty straightforward. It would be more impressive to show efforts to decrease the sim-to-real transfer gap using differentiable simulators.

---

> ### Author Response · Authors · 2022-11-17
> **Author Response**
>
> We thank the reviewer for the comprehensive comments! We detail the response in the following threads. We kindly ask the reviewer to let us know if further clarification or information is needed.
>
> > **Q1**:  How is the data generated? Is it generated by simulations or motion capture systems? How are the parameters of rigid body dynamics such as friction coefficients/restitution coefficients generated?
>
> **A1**: The data is generated from a mix of motion capture and keyframe animations. Specifically, motion clips for the humanoid character are directly borrowed from AMP: "Reference motion clips are collected from a combination of public mocap libraries, custom recorded mocap clips, and artist-authored keyframe animations". Motion clips for the character with sword and shield are from ASE: "a custom motion dataset of 187 motion clips, provided by Reallusion". We follow the setting of DeepMimic in terms of the parameters of rigid body dynamics.
>
> We thank the reviewer for raising the question and have revised the manuscript for better clarity.
>
>
> > **Q2**: Sensitivity to physical parameters change.
>
> **A2**: We definitely agree with the reviewer that this is an important consideration! Thus, we carry out experiments to validate the efficacy of DiffMimic with inaccurate estimates of physical parameters. Briefly speaking, DiffMimic is able to learn the control policy well even though the friction coefficient deviates from the original parameters. We change the friction coefficient to 0.8 and 1.2 of the original value. We refer the reviewer to **Table 6** in the appendix of the updated manuscript.
>
> In addition, we show the robustness of DiffMimic to external force in **Table 5** in the appendix of the updated manuscript. We refer the reviewer to https://diffmimic-demo-main-g7h0i8.streamlit.app/ for visualization.
>
> > **Q3**: Adding adaptivity in teacher forcing is explored in NLP.
>
> **A3**: The adaptive scheme in DiffMimic arises from an important observation that even though the average pose error is seemingly good, the quality of the motion may not be high. We refer the reviewer to Figure 8 for details. In the context of motion mimicking, adaptivity is important to get high-quality motions and is underexplored. It is not immediately clear why adaptivity is required and how adaptivity should be built by observing the average error comparison for adaptive TF and TF.  We are the first to add adaptivity to TF in the motion-mimicking task.
>
>
> > **Q4**: It would be more impressive if the authors could use differentiable physics to remove the assumption of known parameters of rigid body dynamics,
>
> **A4**: Learning system configuration is definitely a promising future direction and one of the motivations behind DiffMimic. For example, the system parameters can be learned through system identification [1,2] with DPS efficiently.
>
> Diffmimic opens up the possibility to adjust the character's weight distribution/actuator force/damping factors along with mimicking. The parameter of the environment and the policy can be learned together.  However, methods for incorporating system identification into the learning loop can be highly task-dependent and may require a separate set of analyses beyond the scope of our contribution of DPS-based motion mimicking. We consider it an important future direction.
>
> [1] Murthy, J. Krishna, et al. "gradSim: Differentiable simulation for system identification and visuomotor control." International Conference on Learning Representations. 2020.
> [2] Le Lidec, Quentin, et al. "Differentiable simulation for physical system identification." IEEE Robotics and Automation Letters 6.2 (2021): 3413-3420.
>
> > **Q5**: One of the important strengths of RL-based methods is that they can be easily transferred onto real robots. However, the policy rollout on real robots is not usually differentiable, which limits the application of the proposed method.
>
> **A5**: We would like to clarify that the policy rollout of DiffMimic during deployment does not require differentiable dynamics. Though the policy is trained via differentiable physics, as a model-free policy, it requires no environment dynamics for deployments.

---

> > ### Comment · Reviewer_yUu6 · 2022-12-06
> > **Thanks for your response!**
> >
> > I thank the authors for their response and their efforts to make this paper better. I would like to raise the score a little bit, but I think the paper has not reached the level of score 8. So I decided to keep my current score.

---

> > > ### Author Response · Authors · 2022-12-06
> > > **Thank you for recognizing our response!**
> > >
> > > Thank you for recognizing our response! We really appreciate your detailed comments and suggestions for improvement. We will include the additional revisions in the final version.

---

### Author Response · Authors · 2022-11-17
**General Response**

We sincerely thank all the reviewers for your constructive feedback and recognition of this work, especially for acknowledging that

* DiffMimic speeds up learning for motion mimicking, an important class of problem.  (Reviewer s4rT, 8KfC)
* DiffMimic is well-supported by extensive experiments on complex and dynamic motions with good results. (Reviewer yUu6, JNig, a2aF)
* DiffMimic presents a new direction toward motion mimicking problems with differentiable physics simulators and shows the correct practice.  (Reviewer JNig, s4rT, a2aF)

We would like to re-emphasize this work's novelty and technical contributions:
* DiffMimic is the first to utilize differentiable physics simulators for motion mimicking. We contribute a few practical techniques to tackle the local optimal challenge in DPS.
* DiffMimic significantly improves the sample efficiency and training time compared to RL-based methods like DeepMimic with comparable or better motion quality, hence making motion mimicking more scalable for many downstream tasks.
* We release DiffMimic as a standard benchmark. The environments can hopefully bring additional future research opportunities to the community, particularly research related to motion generation/mimicking based on differentiable physics.

***

We have revised our manuscript to include the following changes according to all the reviewers’ insightful comments. Note that all the polishments on the main submission and supplemental document are highlighted with **blue** text color for better visualization.

Inclusion of new experiments:
* Experiment on a new character, Ant. We show that DiffMimic generalizes to new characters.
* Experiment on a new skill, 360-degree Jump. We reveal that DiffMimic is able to mimic some challenging motions that RL-based methods fail to.
* Experiment on a new training strategy, DiffMimic with RSI, which requires 90% fewer samples than DeepMimic and has better accuracy. We show that DiffMimic is a promising direction and has great potential when combined with better training recipes.

Inclusion of more analysis:
* Analysis on system sensitivity. We show that DiffMimic performs consistently when system parameters like the friction coefficient are inaccurately estimated.
* Analysis on perturbation robustness. We show that DiffMimic learns policies that are robust to external pushes during rollout.
* Analysis on motion quality. We use a pose absurdity metric to quantify the motion quality improvement of Demonstration Replay.

Inclusion of more discussions:
* More discussions on DeepMimic reward. We compare DeepMimic reward with our loss function and show that ours has significantly fewer hyper-parameters to tune.
* More discussions on some related works. We present a detailed discussion on SHAC, PODS, SuperTrack, and ScaDiver.

Minor changes:
* We fix the arrangement in Fig 7. for better clarity.
* We fix the missing reference in Related Works.
* We show the performance change as a fraction of the DeepMimic samples in Table 2.

Please do not hesitate to let us know of any additional comments on the manuscript or the changes.

---

### Author Response · Authors · 2022-11-22
**Look forward to your reply!**

Thank you again for your valuable time and insightful comments! We have tried to address your concerns in the updated manuscript and our rebuttal text, and we sincerely look forward to your reply. Any pointers to where our revision and rebuttal are lacking would be highly appreciated.

We would also be open to engaging in a discussion to address any other concerns that are currently affecting your review.
Thank you for your time!

---

### Decision · Program_Chairs · 2023-01-20

**Decision:**

Accept: poster

**Justification For Why Not Higher Score:**

1. The motion quality of the learned motions still has some room to improve.
2.  Sim2real is uncertain.

**Justification For Why Not Lower Score:**

Novel direction for  Motion Mimicking.

**Metareview: Summary, Strengths And Weaknesses:**

Five experts reviewed this paper with mixed scores (4 accept, 1 borderline reject). AC does feel that this work makes interesting contributions by introducing a new direction for motion mimicking using differentiable physics. The reviewers did raise some valuable concerns.  The authors are encouraged to make the necessary changes and include the missing references in the final version.

**Note From Pc:**

if the above contains the word "oral" or "spotlight" please see: "oral" presentation means -> notable-top-5% and "spotlight" means -> notable-top-25%. As stated in our emails, we are disassociating presentation type from AC recommendations